# Development of a cell-free screening assay for the identification of direct PERK activators

**Márcia F. D. Costa**[1,2], **Günter U. Höglinger**[1,2,3]* *, **Thomas W. Rösler**[1,4]

1 Department of Translational Neurodegeneration, German Center for Neurodegenerative Diseases, Munich, Germany, 2 Department of Neurology, Hannover Medical School, Hannover, Germany, 3 Department of Neurology, Ludwig-Maximilians University, Munich, Germany, 4 Department of Neurology, School of Medicine, Technical University of Munich, Munich, Germany

☯ These authors contributed equally to this work.
* Guenter.Hoeglinger@DZNE.de

**Data Availability Statement:** The authors confirm that all data underlying the findings are fully available without restriction. All relevant data are within the paper and its Supporting information files.

## Abstract

The activation of the unfolded protein response, particularly via the PERK pathway, has been suggested as a promising therapeutic approach in tauopathies, a group of neurodegenerative disorders characterized by the abnormal phosphorylation and aggregation of tau protein. So far, a shortage of available direct PERK activators has been limiting the progresses in this field. Our study aimed at the development of a cell-free screening assay enabling the detection of novel direct PERK activators. By applying the catalytic domain of recombinant human PERK, we initially determined ideal conditions of the kinase assay reaction, including parameters such as optimal kinase concentration, temperature, and reaction time. Instead of using PERK's natural substrate proteins, eIF2α and NRF2, we applied SMAD3 as phosphorylation-accepting protein and successfully detected cell-free PERK activation and inhibition by selected modulators (e.g., calcineurin-B, GSK2606414). The developed assay revealed to be sufficiently stable and robust to assess an activating $EC_{50}$-value. Additionally, our results suggested that PERK activation may take place independent of the active site which can be blocked by a kinase inhibitor. Finally, we confirmed the applicability of the assay by measuring PERK activation by MK-28, a recently described PERK activator. Overall, our data show that a cell-free luciferase-based assay with the recombinant human PERK kinase domain and SMAD3 as substrate protein is capable of detecting PERK activation, which enables to screen large compound libraries for direct PERK activators, in a high-throughput-based approach. These activators will be useful for deepening our understanding of the PERK signaling pathway, and may also lead to the identification of new therapeutic drug candidates for neurodegenerative tauopathies.

## Introduction

Tauopathies are a heterogeneous group of neurodegenerative disorders characterized by an abnormal phosphorylation and aggregation of the microtubule-associated protein tau with a progressive accumulation of tau aggregates in neurons [1]. In recent years, there has been

**Funding:** This project was supported by the following grants attributed to G.U.H.: German Research Foundation under Germany's excellence strategy within the framework of Hannover cluster RESIST (Grant number: EXC 2155 – project number 39087428) (https://www.dfg.de); Lower Saxony Ministry for Science and Art: REBIRTH – Research Center for Translational Regenerative Medicine (Grant number: MWK, ZN3440.TP) (https://www.mwk.niedersachsen.de), and further supported by the Volkswagen Foundation (Niedersächsisches Vorab) (https://www. volkswagenstiftung.de) and Petermax-Müller Foundation (Etiology and Therapy of Synucleinopathies and Tauopathies). There was no additional external funding received for this study. The funders had no role in study design, data collection and analysis, decision to publish, or preparation of the manuscript.

**Competing interests:** The authors have declared that no competing interests exist.

growing evidences that activation of the PERK (pancreatic endoplasmic reticulum kinase) pathway, a major branch of the UPR (unfolded protein response) involved in restoring protein homeostasis, may be a promising therapeutic target in the context of tauopathies [2–4].

PERK is an ER (endoplasmic reticulum) transmembrane protein with an ER lumen regulatory domain and a cytoplasmic serine/threonine kinase domain [5]. Under non-stressed conditions, PERK is kept at an inactive state by the chaperone BiP (binding immunoglobulin protein) [6]. Accumulation of unfolded or misfolded proteins or other factors, e.g., changes in calcium homeostasis, trigger ER stress and lead to BiP dissociation and activation of the PERK pathway [7–9]. Although the exact mechanism for PERK activation is still not fully understood, accumulating evidence suggests that PERK is activated upon conformational change and oligomerization [5, 7, 10]. The first oligomeric structures described were interdimers, proposed to contribute to PERK trans-autophosphorylation at residue Thr980, consequently stabilizing the PERK activation loop, a structural component which is important for the interaction of PERK and its natural substrate proteins, e.g., eIF2α [5]. More recently, it was shown that the PERK luminal domain in a transient tetrameric state appears to be crucial for an efficient signal transduction [10]. Activation of the PERK pathway ultimately leads to the phosphorylation of its downstream effectors—eIF2α and NRF2. Phosphorylation of eIF2α reduces the flux of protein in the ER by causing translation arrest, alleviating ER protein stress [11]. Furthermore, PERK activation is known to increase cell survival, particularly via the NRF2-pathway, which induces the transcription of antioxidative genes, helping cells to cope with UPR-induced oxidative stress [12].

So far, several *in vitro* and *in vivo* studies have demonstrated that modulation of the PERK pathway is a promising therapeutic approach in tauopathies. Studies with cells carrying PERK risk alleles from patients with tauopathy concluded that enhanced PERK signaling can decrease vulnerability to ER stress-associated damage [13]. Another study revealed that indirect activation of the PERK pathway by inhibition of its repressors, led to lower levels of tau accumulation [4]. In a previous study by our group, we showed that an activator of the PERK pathway was protective in chemical- and viral-induced tauopathy cell models and *in vivo* [2]. Currently, only very few compounds and proteins have been reported as direct PERK activators. However, the majority of these activators have been identified using cell-based assays [14–18]. A major disadvantage of cell-based assays in discovering protein activators is that the identified compounds may lead to a non-specific indirect activation of PERK, e.g., by influencing a cascade of other cellular mechanisms that finally lead to PERK activation. To identify specific activators that directly interact with PERK is therefore rather unlikely in cell-based assays. The UPR is a highly complex mechanism, which sensitively responds to cellular stress. Any test compound causing stress to the cell is likely to activate PERK, not directly, but indirectly. A lack of direct PERK activators represents a limitation in developing novel therapies for tauopathies in this context. Furthermore, specific activators may enable deepened research in understanding PERK signaling. Therefore, the present work aimed at the development of a robust cell-free assay with a setup that can easily be transferred to HTS (high-throughput screening), allowing the identification of novel direct activators of PERK.

## Materials and methods

### Proteins, chemicals, and kits

Recombinant human PERK (EIF2AK3, active catalytic subunit of PERK protein) (E11-11G) was obtained from SignalChem Biotech (Richmond, Canada). Batch-to-batch characterization of PERK kinase specific activity was provided by the manufacture, using their established radioactive activity assay protocol. Recombinant full-length human SMAD3 (S12-30G) was

also purchased from SignalChem Biotech. The natural PERK substrates, full-length recombinant human eIF2A (ab95932) and Nrf2 (ab202153), and full-length recombinant human calcineurin-B (ab103786) were obtained from Abcam (Cambridge, UK). Modifiers of the PERK pathway, GSK2606414 (HY-18072), CCT020312 (HY-119240), TUDCA (tauroursodeoxycholic acid; HY-19696), resveratrol (HY-16561), metformin (HY-B0627) and compound MK-28 (HY-137207) were obtained from MedChemExpress (New Jersey, USA). Compound A, described in Xie et al., 2015 [14], was kindly provided by Orion Pharma (Espoo, Finland). For the measurement of ATP consumption, we used as basic platform the ADP-Glo™ kinase assay (V6930, Promega, Wisconsin, USA).

## PERK titration curve

The measurements for the PERK titration were carried out in white 384-well microplates (bio-one 384-well small volume™ microplate, Greiner, Kremsmünster, Austria), in triplicates. All solutions were prepared in kinase buffer A, consisting of 40 mM TRIS, 20 mM $MgCl_2$, 0.1 mg/mL BSA, and 50 μM DTT adjusted to pH = 7.5. In protein low binding tubes (LoBind®, Eppendorf, Hamburg, Germany) a two-fold serial dilution of PERK was prepared, ranging from 1.4 nM (0.2 ng/μL) to 348 nM (40 ng/μL). To the different concentrations of PERK solutions, as well as to the background control (i.e., a condition which contained no PERK), the substrate protein SMAD3 and ATP were added to a final reaction volume of 5 μL, to achieve final concentrations of 0.26 μM SMAD3 and 50 μM ATP. The mixture was allowed to react for 1 h at RT (room temperature) and the kinase reaction was terminated the by adding 5 μL of ADP-Glo™ reagent (Promega, Fitchburg, USA) and incubating for 40 min, resulting in complete ATP depletion. The ADP generated during the kinase reaction was then converted to ATP by adding 10 μL of kinase detection reagent (Promega) and incubating for another 40 min at RT, protected from light. The produced ATP was simultaneously used by a luciferase/luciferin reaction to produce light which was measured on a FLUOstar® Omega plate reader (BMG LabTech, Ortenberg, Germany). The obtained result was analyzed with the MARS data analysis software (BMG LabTech) and expressed as relative light units (RLUs). For background correction, RLUs detected for the background control were subtracted from the measured values of the different PERK concentrations. The titration curve was obtained by plotting the background-corrected RLUs against their respective PERK concentration.

## Testing different protein substrates

The different kinase reactions were carried out in white 384-well microplates (Greiner), in triplicates. All solutions were prepared in kinase buffer A. To the different protein substrate conditions, PERK protein and ATP, as well as the different protein substrates (SMAD3, Nrf2 or eIF2α), were added to a final reaction volume of 5 μL, to receive final concentrations of 87 nM PERK, 50 μM ATP, and 0.26 μM of the respective protein substrate to be tested. A 5 μL of 50 μM ATP solution was used as background control. For the PERK autophosphorylation control (i.e., a condition which contained no protein substrate), PERK protein and ATP were added to a final reaction volume of 5 μL, to achieve final concentrations of 87 nM PERK and 50 μM ATP. The mixtures reacted for 1 h at RT, followed by incubation with 5 μL of ADP-Glo™ reagent (Promega) for 40 min to terminate the kinase reaction, as mentioned before. The generated ADP during the kinase reaction was converted to ATP and simultaneously consumed in a luciferase/luciferin reaction through a 40 min incubation with 10 μL of kinase detection reagent (Promega) at RT and protected from light. The generated light was measured in RLUs on a FLUOstar® Omega plate reader (BMG LabTech). For background

correction, RLU measured for the background control was subtracted from the measured values of the different protein substrate conditions and PERK autophosphorylation control.

## Optimal reaction temperature

The kinase reaction was carried out in 0.2 mL tubes (Sapphire PCR tubes, Greiner) at RT or at 37 ˚C on a thermal cycler (SensoQuest, Göttingen, Germany), in triplicates. All solutions were prepared in kinase buffer A. To the different testing solutions (control, GSK2606414 and calcineurin-B), PERK protein, substrate protein SMAD3 and ATP were added to a final reaction volume of 5 µL, to achieve final concentrations of 87 nM PERK, 0.26 µM SMAD3, and 50 µM ATP. In the same reaction volume, GSK2606414 and calcineurin-B were added to their respective solutions, at a final concentration of 0.87 µM. For background control, SMAD3 and ATP were added to a final reaction volume of 5 µL, to reach a final concentration of 0.26 µM SMAD3 and 50 µM ATP. After 1 h, the kinase reaction was terminated by adding 5 µL of ADP-Glo™ reagent (Promega) to each reaction tube and let it incubate at RT for 40 min. The generated ADP during the kinase reaction was converted to ATP through a 40 min incubation with 10 µL of kinase detection reagent (Promega) at RT and protected from light. The different kinase reaction solutions were then transferred to a white 384-well (Greiner) for luminescence readout on a FLUOstar® Omega plate reader (BMG LabTech). After correcting the background as previously described, the RLUs values were normalized to basal PERK activity (i.e., control condition in which neither GSK2606414 nor calcineurin-B were added), at the respective reaction temperature (RT or 37 ˚C) and expressed as percentage of relative PERK activity. The signal-to-background ratio was calculated by dividing the RLUs of each condition by the RLU of the background control.

## Optimal reaction time

To determine the optimal reaction time, 87 nM of PERK reacted with 50 µM of ATP and 0.26 µM of SMAD3, in presence of 0.87 µM of GSK2606414 (inhibition curve) or calcineurin-B (activation curve), or kinase buffer A (basal curve). Similar to previous experiments, all solutions were prepared in kinase buffer A, at a final reaction volume of 5 µL, and run in white 384-well microplates (Greiner), in triplicates. The kinase reaction was terminated at different time-points—15, 30, 60, and 120 min—by adding 5 µL of ADP-Glo™ reagent (Promega) and incubating for 40 min at RT. The ATP consumption was measured on a FLUOstar® Omega plate reader (BMG LabTech), after a 40 min incubation with 10 µL of kinase detection reagent (Promega), at RT, protected from light. After correcting the background as before, data was normalized to PERK basal activity (i.e., condition where neither GSK2606414 nor calcineurin-B were added), in order to calculate the percentage of relative PERK activity. The GSK2606414 (inhibition) and calcineurin-B (activation) curves were obtained by plotting the percentage of relative PERK activity against the respective time-point.

## 'Rescue' experiments with calcineurin-B

In protein low binding tubes (LoBind®, Eppendorf), individual solutions of PERK, GSK2606414, calcineurin-B and a combined solution of SMAD3 and ATP were prepared in kinase buffer A, in concentrations four times higher than the final reaction concentrations (i.e., 0.35 µM PERK, 3.48 µM GSK2606414, 3.48 µM calcineurin-B, 1.0 µM SMAD3, and 200 µM ATP). The different experimental conditions were assembled in white 384-well microplates (Greiner), in triplicates. 1.25 µL of 0.35 µM PERK was alternatively pre-incubated with 1.25 µL of 3.48 µM GSK2606414 or calcineurin-B, or kinase buffer A, for 1 h at RT (S1 Fig). After the pre-incubation period, each condition was alternatively 'rescued' with 1.25 µL of

3.48 μM GSK2606414 or calcineurin-B, or kinase buffer A. Finally, 1.25 μL of the combined solution of 1.0 μM SMAD3 and 200 μM ATP was added to each well to start the kinase reaction. The final reaction volume was 5 μL with a final concentration of 87 nM of PERK, 0.26 μM of SMAD3, 50 μM of ATP, and 0.87 μM of GSK2606414 and/or calcineurin-B (except for the control condition that received no PERK modulator). The background control (i.e., a condition which contained no PERK) consisted of a reaction volume of 5 μL containing 0.26 μM SMAD3 and 50 μM ATP. The kinase reaction occurred for 2 h at RT and was terminated by a 40 min incubation with 5 μL of ADP-Glo™ reagent (Promega), at RT. The generated ADP was converted to ATP by a 40 min incubation with 10 μL of kinase detection reagent (Promega, USA), at RT, protected from light. The luminescence signal was measured on a FLUOstar® Omega plate reader (BMG LabTech). Data was corrected to background, as described before, and reported as RLU.

## Determination of calcineurin-B EC$_{50}$

The measurements for the determination of calcineurin-B EC$_{50}$ were carried in white 384-well microplates (Greiner), in triplicates. All solutions were prepared in kinase buffer A. In protein low binding tubes (Eppendorf) a three-fold serial dilution of calcineurin-B was prepared, with the final concentration (i.e., concentration in the well) ranging from 1.2 nM to 24.3 μM. To the different concentrations of calcineurin-B, PERK, substrate protein SMAD3 and ATP were added to a final reaction volume of 5 μL, to achieve final concentrations of 87 nM PERK, 0.26 μM SMAD3, and 50 μM ATP. To determine the basal activity of PERK (i.e., a condition which contained no calcineurin-B), a solution containing 87 nM PERK, 0.26 μM SMAD3, and 50 μM ATP, in a final reaction volume of 5 μL, was prepared. The background control (i.e., a condition which contained no PERK) consisted of a reaction volume of 5 μL containing 0.26 μM SMAD3 and 50 μM ATP. The kinase reaction was carried out at RT for 2 h and terminated by incubation with 5 μL of ADP-Glo™ reagent (Promega) for 40 min, resulting in complete ATP depletion. The ATP consumption was then determined by luminescence measurement on a FLUOstar® Omega plate reader (BMG LabTech), after a 40 min incubation with 10 μL of kinase detection reagent (Promega) at RT and protected from light. The background was corrected as before, and the RLU data were plotted against the log$_{10}$ of calcineurin-B concentrations. The EC$_{50}$ was determined by applying an EC$_{50}$ shift curve with Prism 9 software (GraphPad Software, La Jolla, USA).

## Testing modifiers of PERK activity

Previously reported modifiers of the PERK pathway [3, 14–18] were tested regarding their ability to activate PERK in a cell-free assay. Additionally, bovine serum albumin (BSA) was tested as a negative control. The experiment was carried out in white 384-well microplates (Greiner), in triplicates. The different compounds were incubated at a final concentration of 1.7 μM, corresponding to a molar ratio of 1:20 (i.e., 1 mol of PERK to 20 of testing compound), with 87 nM of PERK, in the presence of 0.26 μM of SMAD3 and 50 μM of ATP, at a final reaction volume of 5 μL. A solution containing 87 nM PERK, 0.26 μM SMAD3, and 50 μM ATP, at a final reaction volume of 5 μL, was prepared for determination of PERK basal activity (i.e., a condition which contained no testing compound). The background control (i.e., a condition which contained no PERK) consisted of a reaction volume of 5 μL containing 0.26 μM SMAD3 and 50 μM ATP. All solutions were prepared in kinase buffer A. The kinase reaction was performed at RT for 2 h and terminated by incubation with 5 μL of ADP-Glo™ reagent (Promega) in each well for 40 min at RT. The ATP consumption was measured as described before, after a 40 min incubation with 10 μL of kinase detection reagent (Promega), at RT and protected

from light. Background correction was applied to all data and data were expressed as percentage of relative PERK activity, as previously described.

## Statistical analysis

Graphical representation and statistical analyses were carried with Prism 9 software (GraphPad). Comparisons between data sets were done by t-test, one-way ANOVA with Dunnett's or Sidak's *post hoc* test, or two-way ANOVA with Tukey's *post hoc* test. A *p*-value $< 0.05$ was considered significant.

## Results

### Setup of a cell-free PERK assay

We started by conceptualizing a screening assay for identification of direct PERK activators, defined as compounds that promote PERK activity by direct interaction with PERK. Cellular screening assays commonly lead to an unclear identification of the compound-target (Fig 1A). In these systems, the successful identification of a test compound commonly starts with the

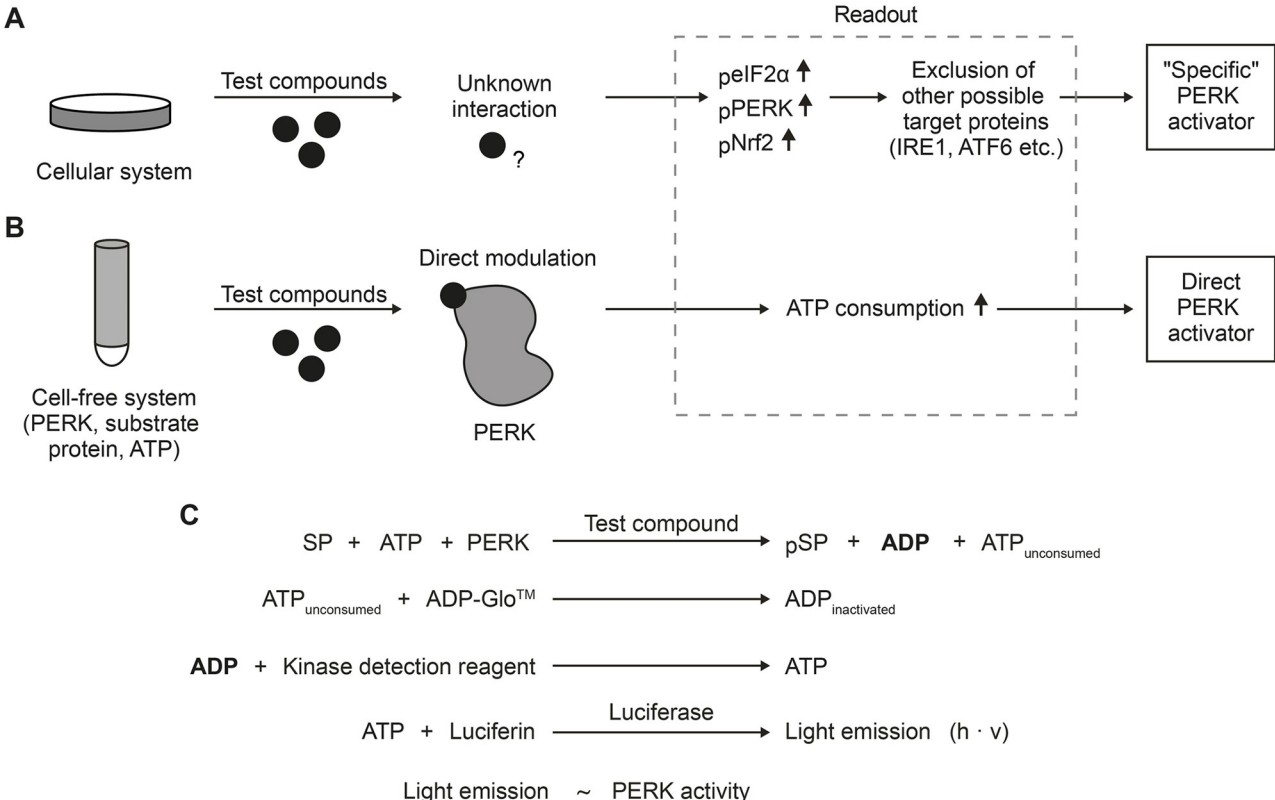

**Fig 1. Concept of the PERK activator screening assay.** A) Cellular screening assays can lead to selection of compounds with unknown target interactions. Given the complexity of the UPR, identified activators are most likely unspecific. However, by measuring distinct activated target proteins or by comparing treated PERK knockout cells with wild-type cells other pathways of the UPR can be excluded finally resulting in the identification of a so-called "specific activator". B) A cell-free screening systems, which is the focus of the present study, allows the detection of compounds that specifically and directly modulate PERK activity. An increased consumption of ATP caused by the test compound, very likely results from the effect of a direct PERK activator. C) Schematic overview of the PERK assay developed in this study, using the ADP-Glo assay as its basis. In total, 4 reaction steps are necessary. At first, PERK phosphorylates the substrate protein (SP) in the presence of ATP. The addition of reagent A terminates the kinase reaction and converts the unconsumed ATP into inactivated ADP. In the third step, reagent B converts the ADP from the first step into ATP. The newly synthesized ATP is then measure by a luciferase/luciferin reaction. The emitted light is directly proportional thus to PERK activity.

measurement of the expression levels of PERK down-stream targets, such as an increased phosphorylation of eIF2α or Nrf2, as well as an increase in the amount of phosphorylated PERK (the activated form of the PERK protein). The compound specificity for PERK activation can then be inferred either by comparing treated PERK knockout cells with wild-type cells or by excluding the activation of other possible target proteins, such as the IRE1 (inositol-requiring enzyme 1) and the ATF6 (activating transcription factor 6), responsible for the activation of two further UPR branches. Considering the complexity of the UPR, cellular screening assays may identify compounds that activate the PERK pathway in an unspecific manner. On the contrary, cell-free screening systems, by restricting the reaction to the target protein, allow the identification of test compounds that specifically and directly modulate PERK activity (Fig 1B).

At present, there is no commercial PERK activator assay available. Because we intended to use a non-radioactive cell-free screening assay for the measurement of PERK activity in the presence of a test compound, we combined the radioactive assay protocol used by SignalChem for batch characterization of PERK activity with the luminescent ADP-Glo assay readout system. By keeping the PERK catalytic subunit and the protein substrate SMAD3 from the original radioactive protocol and applying it to the ADP-Glo setup, as described in detail in the materials and methods section, we established a non-radioactive assay protocol capable of detecting PERK activity in the femtomolar range, having a comparable sensitivity to assays with radioactive readout (Fig 1C). In the first step, PERK reacts in the presence of a test compound, consuming ATP while phosphorylating a substrate protein. This leads to an increase in ADP. In a second step, the addition of the ADP-Glo™ reagent terminates the kinase reaction and converts the unconsumed ATP in inactivated ADP. On the third step, the ADP that was produced during the first step is converted into ATP by addition of the kinase detection reagent. Finally, the amount of ATP is measured through a luciferin/luciferase reaction, and the emitted lighted, proportional to the PERK activity, is quantified.

## Optimizing different parameters of the PERK assay

At first, we wanted to determine the optimal PERK concentration for the assay by measuring a PERK titration curve. Our results showed the development of a linear luminescence signal between 5–20 ng/μL of PERK (Fig 2A). From this relative broad range, we selected 10 ng/μl of PERK, corresponding to 87 nM of PERK, as the optimal PERK concentration for subsequent experiments, since it is located in the middle of the linear range of the PERK titration curve and generated a sufficient signal-to-background ratio, i.e., a signal approximately 35 times higher than the background (Table 1).

We applied SMAD3 as substrate protein in our assay system, because it is successfully used to determine PERK's specific activity in routine protein quality analysis. It furthermore has the advantage to be more cost-efficient, as compared to PERK's natural substrates, such as Nrf2 and eIF2α. However, we also confirmed the suitability of SMAD3 as substrate protein by comparing PERK activity in the presence of different substrate proteins and by assessing possible direct effects on ATP consumption (Fig 2B). Our results showed that none of the tested substrate proteins (NRF2, eIF2α and SMAD3) directly consumed ATP, which would have led to false ATP consumption results in the assay. The consumption of ATP by PERK was higher when the kinase reaction was performed in the presence of a substrate protein, leading to higher RLU values when compared with the RLUs obtained by PERK activity exclusively (PERK autophosphorylation). However, this increase in PERK activity was only statistically significant when using SMAD3 as protein substrate. We also compared the gain of luminescence related to PERK autophosphorylation, which represents the extra amount of ATP

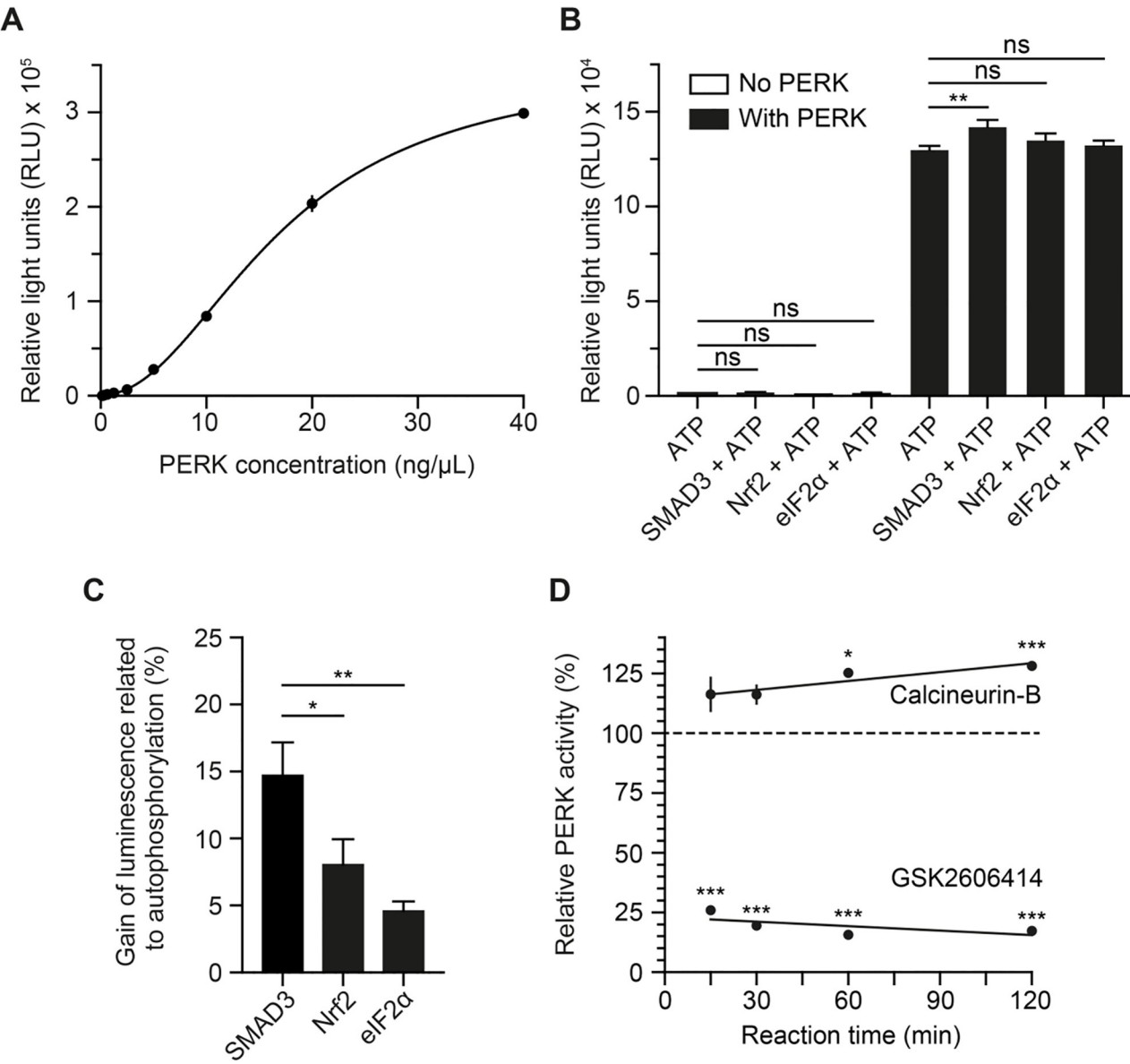

**Fig 2. PERK assay development and validation.** A) PERK titration curve. Graph represents PERK concentration in a two-fold serial dilution against PERK activity expressed as relative light units (RLU), fitted with nonlinear regression curve (Four Parameters Logistic Regression (4PL)). Data plotted as mean ± SEM from 3 independent experiments. B) Testing of different protein substrates. PERK natural substrates, eIF2α and NRF2, and SMAD3 were tested as protein substrates, in absence or presence of PERK. PERK activity was measured as a luminescence signal and expressed as RLU. Data are mean ± SEM from 3 independent experiments. Statistical analysis was 2-way ANOVA with Tukey's *post hoc* test. **$p < 0.01$. C) Performance comparison of tested protein substrates. Graph show the gain of luminescence related to PERK autophosphorylation (no substrate protein). Data are mean ± SEM from 3 independent experiments. Statistical analysis was 1-way ANOVA with Dunnett's post hoc test. *$p < 0.05$ and **$p < 0.01$. D) Testing an inhibitor and an activator. Effects of calcineurin B and GSK2606414 on PERK activity at different reaction times. Measured RLUs were normalized to basal PERK activity (control condition), which is set as 100% (dashed line). Data are mean ± SEM from 3 independent experiments. Statistical analysis was multiple *t*-test between control condition and GSK2606414 or calcineurin-B condition, respectively. *$p < 0.05$ and ***$p < 0.001$.

consumed by PERK when reacting in the presence of the different substrate proteins (SMAD3, NRF2 or eIF2α) compared with the amount of ATP consumed in the absence of a substrate protein (PERK autophosphorylation) (Fig 2C). The highest gain of luminescence related to PERK autophosphorylation was observed when using SMAD3 as substrate protein, leading to

**Table 1. Signal-to-background ration values from PERK titration curve.**

| PERK concentration (ng/µL) | Signal-to-background ratio (mean ± SEM) |
|---|---|
| 40 | 118.2 ± 13.1 |
| 20 | 79.0 ± 5.4 |
| 10 | 35.2 ± 6.5 |
| 5 | 12.9 ± 3.6 |
| 2.5 | 3.6 ± 1.1 |
| 1.3 | 2.3 ± 0.4 |
| 0.6 | 1.6 ± 0.0 |
| 0.3 | 1.2 ± 0.1 |
| 0.2 | 1.1 ± 0.0 |

an almost 15% stronger luminescence signal. To a lesser extent, both of PERK's natural substrates, NRF2 and eIF2α, also contributed to a gain of luminescence, indicating that they acted as phosphorylation-accepting proteins, too. Overall, SMAD3 proved to be a suitable substrate protein for our assay system, also, because it was slightly more prone to PERK phosphorylation than PERK's natural substrate proteins, NRF2 and eIF2α.

Next, we wanted to explore the influence of temperature on the kinase reaction. For that, we perform the kinase reaction in the presence of calcineurin-B, a previously described PERK activator [19], and GSK2606414, a strong inhibitor of PERK [20], at RT and at 37 ºC. We observed strong significant differences in PERK activity between the experimental groups. GSK2606414 led to a strong and statistically significant inhibition of PERK activity when compared with the control (p < 0.001), both at RT and at 37 ˚C (Table 2). However, calcineurin-B revealed a significant increase in PERK activity (p < 0.05) only when the kinase reaction was performed at RT (Table 2). Interestingly, when reacting at 37 ˚C we mainly observed a reduction of the signal-to-background ratio for all the testing conditions (Table 2). Therefore, RT was chosen as a suitable temperature for the kinase reaction, allowing for the detection of PERK activating effects with a given test compound, with higher signal-to-background ratios.

Finally, we investigated PERK activation and inhibition over time to determine the minimum kinase reaction time necessary to detect significant differences in ATP consumption between the different testing conditions (PERK (basal activity), calcineurin-B-activated PERK and GSK2606414-inhibited PERK). GSK2606414 quickly and strongly inhibited PERK, significantly reducing its activity to approximately 25% of the basal PERK activity (absence of a PERK modulator) (Fig 2D). The inhibition was not markedly intensified with a prolonged reaction time. Calcineurin-B started to show signs of activation quickly, although the increase in PERK activity was only significant when the kinase reaction developed over 60 min. At the

**Table 2. Testing PERK inhibition and activation at RT and 37 ˚C.**

| | RT | | 37 ˚C | |
|---|---|---|---|---|
| | % of PERK activity (mean ± SEM; n = 3) | Signal-to-background ratio (mean ± SEM; n = 3) | % of PERK activity (mean ± SEM; n = 3) | Signal-to-background ratio (mean ± SEM; n = 3) |
| Control | 100.0 ± 1.8 | 6.0 ± 0.5 | 100.0 ± 1.9 | 2.6 ± 0.2 |
| GSK2606414 | 14.5 ± 1.6[a] | 1.2 ± 0.2 | 24.9 ± 2.3 [a] | 0.9 ± 0.2 |
| Calcineurin-B | 106.1 ± 1.7 [a] | 6.3 ± 0.6 | 99.5 ± 4.0 [a] | 2.6 ± 0.3 |

[a]Activity values related to control.

longest reaction time (120 min), calcineurin-B increased PERK activity up to 128% (Fig 2D), measurable by a strong statistical difference when compared with PERK basal activity. Therefore, we defined 120 min as the optimal reaction time for the kinase reaction to detect an increase in ATP consumption making it possible to assess PERK activation.

## Calcineurin-B rescuing effects and calcineurin-B EC$_{50}$

First, we wanted to confirm that the increased ATP consumption which we had observed when PERK reacted in the presence of calcineurin-B was not due to a direct consumption of ATP by calcineurin-B nor due to the possibility that calcineurin-B mimics a PERK substrate, accepting additional phosphorylation. In the absence of PERK, we did not observe any considerable consumption of ATP (Fig 3A). The measured RLU values did not differ from the RLU values measured when reacting only the substrate protein with ATP, indicating that calcineurin-B does not directly consume ATP. We also observed that in the presence of PERK, but in the absence of the substrate protein, calcineurin-B did not contribute to a higher PERK activity when compared to the basal activity of PERK autophosphorylation of PERK (Fig 3A). This suggested that calcineurin-B does not act as a substrate protein and its effect on PERK activity is due to an increased phosphorylation of a substrate protein rather than a result of an increased PERK autophosphorylation.

   We wanted to further understand the mechanism of action of calcineurin-B on PERK activation, by testing its ability to rescue or prevent PERK inhibition by GSK2606414 in our assay. As observed previously, both PERK modulators, calcineurin-B and GSK2606414 respectively, activated and inhibited PERK when compared to the control condition (PERK basal activity). Interestingly, we observed that pre-incubation with calcineurin-B, one hour before starting the kinase reaction, led to a significantly higher RLU value when compared with pre-incubation with only kinase buffer A (Fig 3B). Pre-incubation with GSK2606414 did not lead to a stronger inhibition of PERK. Additionally, we found that pre-incubation with calcineurin-B partially prevented the strong inhibition with GSK2606414. Similarly, calcineurin-B was able to partially 'rescue' the inhibition caused by a pre-incubation with GSK2606414. The incubation with calcineurin-B, before or after adding GSK2606414, resulted in a similar PERK activity. Our findings strongly suggest that calcineurin-B's site of action differs from the site of action of the inhibitor GSK2606414 and activates PERK by a mechanism independent of the active site which is blocked by GSK2606414.

   Additionally, we tested the capacity of our assay to detect PERK activation by determining the EC$_{50}$ of calcineurin-B. For that, we measured the consumption of ATP by PERK when incubated with increased concentrations of calcineurin-B, using a three-fold serial dilution. We observed a tendential PERK activation starting at the nanomolar range (900 nM), with the maximum activity occurring at micromolar scale (15.15 μM) (Fig 3C). We obtained an EC$_{50}$ of 2.72 μM for calcineurin-B.

## Testing modifiers of the PERK pathway in the cell-free assay

After setting all parameters of our assay and confirming its ability to detect PERK activation and its suitability to determine an activating EC$_{50}$, we decided to run a small pilot screening with six selected compounds that have been reported in the literature as activators of the PERK pathway in cellular screening assays [3, 14–18]. The compounds were previously tested in different molar ratios and a molar ratio of 1:20 (i.e., one mole of PERK to 20 moles of test compound) revealed to be the best molar ratio to detect PERK activation. In our assay, the compound MK-28 significantly activated PERK when compared to control (PERK basal activity) (Table 3). However, a direct ATP consumption by MK-28 could be excluded (S2 Fig). For

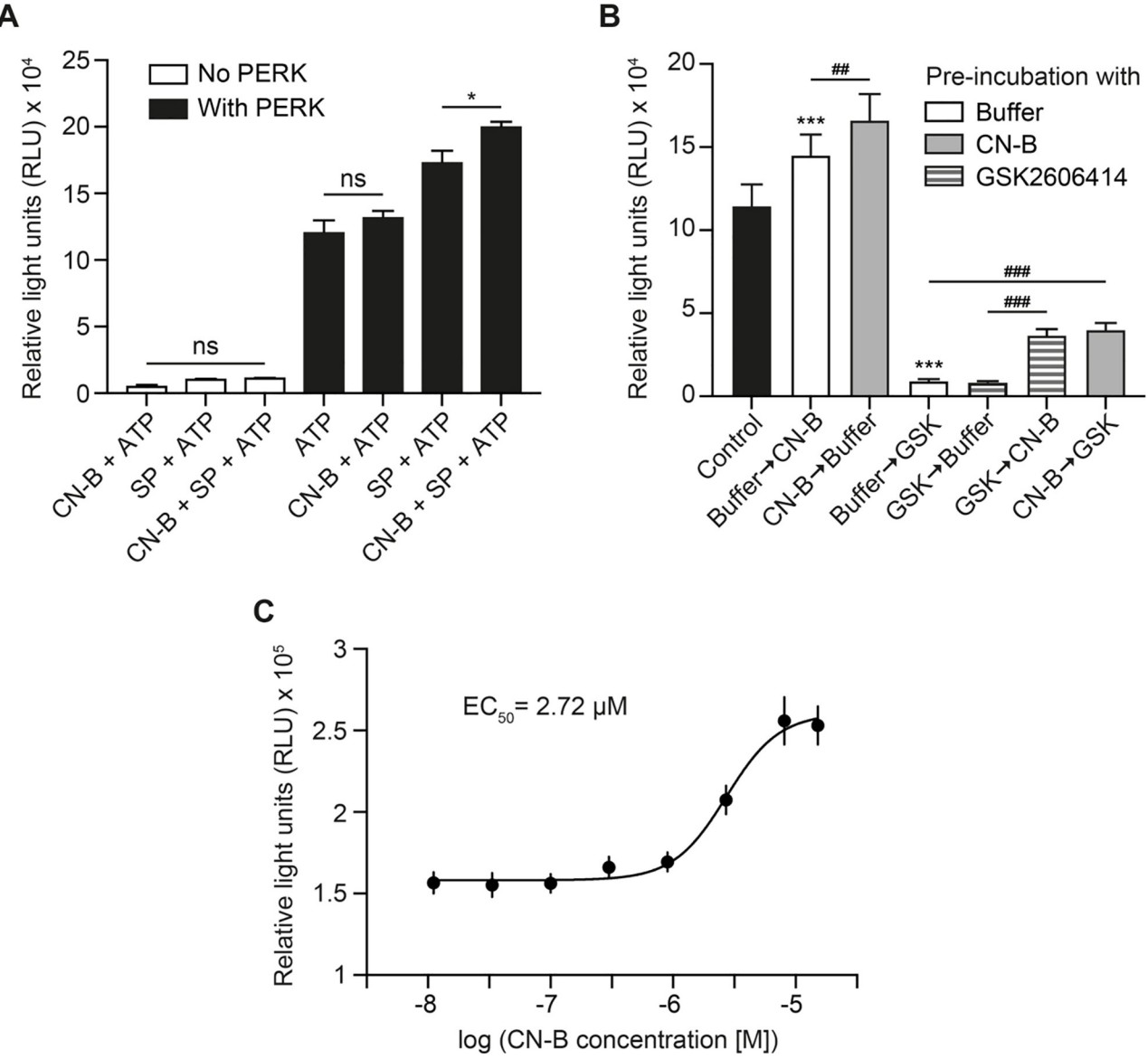

**Fig 3. Characterization of calcineurin-B as a direct activator of PERK in the developed assay.** A) Investigation of the direct effects of calcineurin-B on ATP consumption. Effects of calcineurin-B on ATP consumption were measured in absence or presence of PERK and expressed as RLU. Data are mean ± SEM from 4 technical replicates. Statistical analysis was 2-way ANOVA with Tukey's *post hoc* test. $^*p < 0.05$. B) Influence of calcineurin-B on PERK inhibition caused by GSK2606414. The effects of calcineurin-B and GSK2606414 on PERK activity were evaluated upon pre-incubation with calcineurin B, GSK2606414 or buffer. PERK activity was measured and expressed as RLU. Data are mean ± SEM from 3 independent experiments. Statistical analysis was 1-way ANOVA with Tukey's *post hoc* test. $^{***}p < 0.001$ vs control, $^{##}p < 0.01$ and $^{###}p < 0.001$. C) $EC_{50}$ of calcineurin-B determined by a three-fold serial dilution. Graph represents the log of calcineurin-B concentration (from 11.1 nM to 15.2 μM) against PERK activity measured as RLU, fitted to a nonlinear regression curve ($EC_{50}$ shift). Data are mean ± SEM from 3 independent experiments.

the other test compounds, we could not confirm a statistically significant direct activation of PERK with our assay system.

## Discussion

Activation of the PERK pathway is crucial for protein homeostasis and cell survival [11, 12], and offers a potential therapeutic approach particularly for tauopathies [2–4]. However, a lack

**Table 3. Testing different modifiers of the PERK pathway.**

|  | Relative PERK activity in % [mean ± SEM; n = 3] | p-value |
|---|---|---|
| Control | 100.0 ± 0.7 | N/A |
| CCT020312 | 105.3 ± 1.3 [a] | p = 0.9 |
| Compound A | 102.5 ± 3.5 [a] | p = 1.0 |
| Metformin | 100.9 ± 4.2 [a] | p > 1.0 |
| MK-28 | 114.5 ± 2.8 [a] | **p < 0.001** |
| Resveratrol | 101.9 ± 3.3 [a] | p = 1.0 |
| TUDCA | 103.7 ± 2.0 [a] | p = 1.0 |

[a] Activity values related to control.

of available compounds that directly activate PERK limits the research in this field. Developing a cell-free screening assay for kinase activators is highly challenging compared to assays for inhibitors. Kinase activators act in mechanistically diverse ways, whereas inhibitors commonly bind to the active site of the target protein, blocking its catalytic activity. Additionally, when kinase activity is enhanced by an activator, the reaction rate increases, leading to a faster consumption of the reactants. To prevent limiting the reaction due to total consumption of its reactants, larger quantities of ATP and substrate protein are necessary when screening for kinase activators [21]. This makes the overall cost of a screening assay for kinase activators higher than for kinase inhibitors. At present, the majority of described PERK activators were identified in cellular studies [3, 14–18, 22]. However, cellular systems do not allow to automatically conclude a direct modulation of PERK by the test compound, because the test compound may also trigger endoplasmic stress and might activate the UPR, including PERK, in an unspecific manner. For this reason, cell-free systems are an attractive alternative, because the reaction is restricted to the target protein, allowing the identification of test compounds that specifically and directly modulate the target. Currently, there is no cell-free screening kit for PERK activators available. Existing radioactive methodologies to measure PERK activity in the presence of modulators require special infrastructures which are not present in every laboratorial facility. Therefore, we aimed to design a cell-free assay capable of detecting direct PERK activation very sensitively by combining a PERK assay protocol with a radioactive readout with the advantages of the sensitive luminescent ADP-Glo assay readout system. Using the catalytic subunit of recombinant human PERK and the human SMAD3 as substrate protein, we started by determining the ideal conditions of the kinase assay reaction, such as optimal kinase concentration, temperature and reaction time. We found that SMAD3, a protein substrate involved in the transduction of the TGF-beta pathway [23], was more prone to PERK phosphorylation than PERK's natural substrates, such as NRF2 and eIF2α, making it a suitable and cost-efficient alternative in future HTS applications. With our assay setup, we successfully detected a fast and strong PERK inhibition with GSK2606414, a known PERK inhibitor [20, 24, 25], but more interestingly, we were able to measure as well a robust activation with calcineurin-B, the regulatory subunit of a calcium/calmodulin-dependent phosphatase that contributes to ER homeostasis [19, 26]. Calcineurin has been described to interact and promote PERK activation *in vitro* in radioactive kinase assays [19, 26]. Similarly, we detected a consistent PERK activation by calcineurin-B with our assay setup, and showed that the developed assay is sufficiently stable and robust to assess an activating $EC_{50}$ value. Moreover, our experiments suggest that PERK activation by calcineurin-B involves a mechanism that does not affect the catalytic site. Interestingly, we observed that calcineurin-B does not favor PERK autophosphorylation and only activates PERK in the presence of a substrate protein. In line with

evidences of a weak PERK-calcineurin B-interaction reported by others [19], it is possible to speculate that calcineurin-B might increase PERK activity not by directly interacting with PERK's catalytic site, but by favoring protein-protein interactions between PERK and its protein substrate, favoring phosphorylation of the protein substrate with consequent consumption of ATP, translating into a higher PERK activity. In fact, improving stability of kinase-substrate interaction, by targeting structural features prone to disruption, is a possible mechanism of action of kinase activators [27]. Finally, when testing the assay applicability for the detection of direct PERK activators, we were able to confirm PERK activation by the compound MK-28, a recently described activator of PERK [3]. For many other compounds, which have been described in the literature to activate PERK in cellular systems [3, 14–18], we were not able to detect PERK activating effects in our assay. This lack of effect may be explainable with an indirect activation mechanism of the PERK pathway by these compounds, which is only measurable in cellular systems. It is known that any stimuli susceptible to trigger ER stress can activate the PERK pathway in an unspecific manner as well. For instance, resveratrol, which has been described as a PERK activator, is known for interfering with calcium-homeostasis, thus triggering ER stress without necessarily activating PERK directly [28]. In line with this, our data suggests that resveratrol is not a direct activator of PERK. However, it has to be noted that we used the catalytic domain of PERK in our experiments, as the full-length active PERK protein is so far not available. Further variations which differ from biological conditions included the usage of SMAD3 instead of the PERK's natural substrates, as well as the reaction temperature. Although unlikely, these differences may lead to the selection of positive molecule hits with the developed assay that later may fail to activate the PERK pathway under *in vivo* conditions. Thus, although the described cell-free assay provides a useful screening method for PERK activators due to its simplicity, moderate costs and relatively easy up-scale, identified positive hits will require further confirmation and characterization in biological systems.

During the experiments, we also observed small fluctuations in ATP consumption for the majority of the tested compounds. However, those fluctuations were also seen for the control protein BSA, suggesting them to be unspecific. For future HTS application, the establishment of an activation threshold may help to overcome these unspecific signal variations and avoid false hits.

## Conclusion

Overall, our data suggest that a cell-free screening assay using the catalytic domain of recombinant human PERK and SMAD3 as protein substrate is capable of detecting PERK activation and may be applied in HTS to discover novel direct PERK activators. The activation of the PERK pathway, a major branch of the UPR, seems to be a promising therapeutic approach in tauopathies and other neurodegenerative disorders. Novel direct PERK activators will be useful to deepen our understanding of the PERK signaling pathway and helpful in identifying new therapeutic drug candidates.

## Supporting information

**S1 Fig. Diagram of the different experimental conditions tested in the 'rescue' experiments with calcineurin-B (CN-B).** PERK was pre-incubated with CN-B, GSK2606414 or buffer. After the pre-incubation, PERK was alternatively 'rescue' with CN-B, GSK2606414 or buffer, as indicated in the diagram. Finally, SMAD3 and ATP were added to start the kinase reaction. The ATP consumption was measured as detailed in the materials and methods section. (TIF)

**S2 Fig. Assessing direct effects of MK-28 on ATP consumption.** MK-28 effect on ATP consumption was measured in absence and presence of PERK and expressed as RLU. Data are mean ± SEM from 3 technical replicates. Statistical analysis was Student's *t*-test. $^*p < 0.05$ and $^{***}p < 0.001$.
(TIF)

## Acknowledgments

The authors would like to thank Luciana Fernandes for helping with the preliminary setup of experiments and to Mojtaba Nemati for proofreading the manuscript.

## Author Contributions

**Conceptualization:** Günter U. Höglinger, Thomas W. Rösler.

**Data curation:** Márcia F. D. Costa.

**Formal analysis:** Márcia F. D. Costa, Thomas W. Rösler.

**Funding acquisition:** Günter U. Höglinger.

**Investigation:** Márcia F. D. Costa.

**Methodology:** Márcia F. D. Costa.

**Supervision:** Günter U. Höglinger, Thomas W. Rösler.

**Visualization:** Márcia F. D. Costa, Thomas W. Rösler.

**Writing – original draft:** Márcia F. D. Costa.

**Writing – review & editing:** Günter U. Höglinger, Thomas W. Rösler.

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
