## [Decision Letter · Decision Letter 0]

2 Feb 2023

PONE-D-22-33273Development of a cell-free screening assay for the identification of direct PERK activatorsPLOS ONE

Dear Dr. Costa,

Thank you for submitting your manuscript to PLOS ONE. After careful consideration, we feel that it has merit but does not fully meet PLOS ONE’s publication criteria as it currently stands. Therefore, we invite you to submit a revised version of the manuscript that addresses the points raised during the review process.Your manuscript was found to be interesting. However, reviewer suggests that it requires more clarifications. Please discuss limitations of the assays performed and scope of the study. There are several other comments that need your attention. Please submit your revised manuscript by Mar 19 2023 11:59PM. If you will need more time than this to complete your revisions, please reply to this message or contact the journal office at plosone@plos.org. Please include the following items when submitting your revised manuscript:A rebuttal letter that responds to each point raised by the academic editor and reviewer(s). You should upload this letter as a separate file labeled 'Response to Reviewers'.A marked-up copy of your manuscript that highlights changes made to the original version. You should upload this as a separate file labeled 'Revised Manuscript with Track Changes'.An unmarked version of your revised paper without tracked changes. You should upload this as a separate file labeled 'Manuscript'.If applicable, we recommend that you deposit your laboratory protocols in protocols.io to enhance the reproducibility of your results. Protocols.io assigns your protocol its own identifier (DOI) so that it can be cited independently in the future. For instructions see: https://journals.plos.org/plosone/s/submission-guidelines#loc-laboratory-protocols. Additionally, PLOS ONE offers an option for publishing peer-reviewed Lab Protocol articles, which describe protocols hosted on protocols.io. Read more information on sharing protocols at https://plos.org/protocols?utm_medium=editorial-email&utm_source=authorletters&utm_campaign=protocols.

We look forward to receiving your revised manuscript.

Kind regards,

Hemant K. Paudel

Academic Editor

PLOS ONE

Journal Requirements:

"This project was supported by the German Research Foundation under Germany’s excellence strategy within the framework of Hannover cluster RESIST (EXC 2155 – project number 39087428); by the Lower Saxony Ministry for Science and Art (MWK, ZN3440.TP): REBIRTH – Research Center for Translational Regenerative Medicine; Volkswagen Foundation (Niedersächsisches Vorab) and Petermax-Müller Foundation (Etiology and Therapy of Synucleinopathies and Tauopathies)."

"This project was supported by the German Research Foundation under Germany’s excellence strategy within the framework of Hannover cluster RESIST (EXC 2155 – project number 39087428); by the Lower Saxony Ministry for Science and Art (MWK, ZN3440.TP): REBIRTH – Research Center for Translational Regenerative Medicine; Volkswagen Foundation (Niedersächsisches Vorab) and Petermax-Müller Foundation (Etiology and Therapy of Synucleinopathies and Tauopathies)."

Reviewers' comments:

Reviewer's Responses to Questions

**Comments to the Author**

1. Is the manuscript technically sound, and do the data support the conclusions?

Reviewer #1: Yes

2. Has the statistical analysis been performed appropriately and rigorously? 

Reviewer #1: Yes

3. Have the authors made all data underlying the findings in their manuscript fully available?

Reviewer #1: Yes

4. Is the manuscript presented in an intelligible fashion and written in standard English?

Reviewer #1: Yes

5. Review Comments to the Author

Reviewer #1: This manuscript reports the development of a screening assay for the identification of direct PERK activators. The development of this type of cell-free assay is timely, as there is evidence for neuroprotective effects by inducing direct PERK activation. The following concerns must be addressed.

1) Fig. 2B. There is practically no increase in the signal obtained in the assay using the natural PERK substrates. Even the increase with the unnatural substrate SMAD3 is very small, although in this case it gave statistical significance. These results point to a lack of robustness of the assay and put in doubt its validity to measure the natural activity of PERK. Although it might still be useful to screen for PERK activators, these limitations of the study should be discussed in the text.

2) It is unclear why SMAD3 was chosen as a substrate, as it is not a natural substrate of PERK.

3) The use of the unnatural substrate SMAD3, of only the catalytic domain of PERK and of a reaction at room temperature instead of 37C, takes the developed assay away from the natural PERK biology. While it might still be useful to screen for PERK activators, this should be commented and discussed as a limitation of the study.

4) The statement in the Abstract about the “…non-availability of direct PERK activators…” as well as at the beginning of the Discussion and in other parts of the text is misleading, as MK-28 has been shown to be a direct PERK activator in a cell-free assay (Ganz el al, 2019) and confirmed now in this manuscript. These statements should be removed or changed.

5) Fig. 1A, and lines 258-9. “…cellular screening assays may identify compounds that activate the PERK pathway in an unspecific manner…” Although this could be true in general, there are ways to test the specificity to PERK, by measuring the effect of the compounds in PERK knockout cells compared to WT cells. This should be added/ commented in the figure and in the text.

6) Lines 259-261 “Contrariwise, cell-free screening systems, by restricting the reaction to the target protein, allow the identification of test compounds that specifically and directly interact with the PERK kinase (Fig 1B).” This statement gives the impression that direct interaction is measured in the manuscript, which is not the case. Direct action on PERK can be inferred but direct interaction has not been measured. The statement should be removed or moderated.

7) Line 277 “At present, there is no commercial PERK activator assay available.” and line 458 “…there is no cell-free screening kit for PERK activators obtainable…” However, there are companies that have cell-free PERK activator assays, and one can make use of their services (see e.g. in Ganz el al 2019). This should be commented in the text.

8) Line 281 “…protein substrate SMAD3 from the original radioactive protocol and applying it…”. It is unclear to what original radioactive protocol the authors are referring to.

9) Lines 410-412 “Our findings strongly suggest that calcineurin-B place of action differs from the inhibitor GSK2606414 and activates PERK by a mechanism independent of the active site which is blocked by GSK2606414.” The authors should speculate in the text what the mechanism of activation could be, based on previous studies on the mechanisms of PERK activation.

10) Although the English is good in general, there are some mistakes in the grammar. (for example in line 373 “…when PERK reacted in the presence of calcineurin-B was neither not due to a direct…”). The text should be edited by a native English speaker.

6. PLOS authors have the option to publish the peer review history of their article (what does this mean?). If published, this will include your full peer review and any attached files.

Reviewer #1: No

---

## [Author Response · Author response to Decision Letter 0]

15 Mar 2023

Editor´s comments:

Comment 1: Please ensure that your manuscript meets PLOS ONE's style requirements, including those for file naming.

Answer: We revised once more our manuscript to ensure it follows the style requirements. The following adjustments were applied throughout the text.

• File names were corrected, including file name of supplemented figure 1 and 2

• Footnote symbols of Table 2 and 3 were corrected

• In line 361, we added a p-value that had accidentally been deleted prior to submission

• We fixed the S1 Fig, S2 Fig and Fig 3 captions. Supporting figures captions were moved to the end of the manuscript, in a section titled “Supporting

• information”

• We corrected the heading in line 537, so that only first word is capitalized

• A first line space was added to mimic the layout of PLOS ONE template

Comment 2: Please provide an amended statement that declares *all* the funding or sources of support received during this study. Please also include the statement “There was no additional external funding received for this study.” in your updated Funding Statement. 

Answer: We would like to provide an amended Financial Disclosure as following:

This project was supported by the following grants attributed to G.U.H.: German Research Foundation under Germany’s excellence strategy within the framework of Hannover cluster RESIST (Grant number: EXC 2155 – project number 39087428) (https://www.dfg.de/en/index.jsp); Lower Saxony Ministry for Science and Art: REBIRTH – Research Center for Translational Regenerative Medicine (Grant number: MWK, ZN3440.TP) (https://www.mwk.niedersachsen.de/startseite/), and further supported by the Volkswagen Foundation (Niedersächsisches Vorab) (https://www.volkswagenstiftung.de/de) and Petermax-Müller Foundation (Etiology and Therapy of Synucleinopathies and Tauopathies) (https://www.porsche-hannover.de/petermax_mueller_gruppe_de_han,602724.html). There was no additional external funding received for this study. The funders had no role in study design, data collection and analysis, decision to publish, or preparation of the manuscript.

Comment 3: Please state what role the funders took in the study. If the funders had no role, please state: "The funders had no role in study design, data collection and analysis, decision to publish, or preparation of the manuscript.". 

Answer: Please see answer to comment 2.

Comment 4: In your Data Availability statement, you have not specified where the minimal data set underlying the results described in your manuscript can be found. We will update your Data Availability statement to reflect the information you provide in your cover letter.

Answer: In our Data Availability we stated that “all data are fully available without restriction” and that “all relevant data are within the manuscript and its Supporting Information files”, as instructed by PLOS ONE submission guidelines.

Comment 5: Please review your reference list to ensure that it is complete and correct. Any changes to the reference list should be mentioned in the rebuttal letter that accompanies your revised manuscript.

Answer: We carefully revised our reference list and excluded the former reference 22 (Guan et al, 2020), cited once in line 460, since this article did not move to peer review, up to now. The reference list and the citations through the text were modified to reflect this change. 

Reviewer´s comments:

Comment 1:Fig. 2B. There is practically no increase in the signal obtained in the assay using the natural PERK substrates. Even the increase with the unnatural substrate SMAD3 is very small, although in this case it gave statistical significance. These results point to a lack of robustness of the assay and put in doubt its validity to measure the natural activity of PERK. Although it might still be useful to screen for PERK activators, these limitations of the study should be discussed in the text.

Answer: Thank you for the comment. As stated by the reviewer, the increase in PERK activity by SMAD3 was small, although statistically significant. However, in our view, this is not necessarily suggestive for a lack of robustness or validity. The aim of this particular experiment was to simply understand if SMAD3, routinely used in the radioactive assay protocol setup by the manufacturer SignalChem, is a suitable protein substrate in our setup. For this reason, we compared PERK´s ATP consumption in presence of SMAD3, or in presence of PERK´s natural substrates (eIF2α and NRF2). Although PERK is capable of autophosphorylation, leading to ATP consumption even in the absence of a protein substrate, an increase in ATP consumption is expected when a substrate protein is present. Indeed, as shown in Fig 2B, we observed an increase in ATP consumption when PERK reacted in presence of any of the tested substrate protein, suggesting that both SMAD3 and PERK´s natural substrates are working as phosphorylation-accepting proteins. Moreover, the increase in ATP consumption (suggestive of a gain in PERK activity) was higher and statistically significant for SMAD3, thus reinforcing the cost-efficiency of SMAD3 compared to PERK´s natural substrates. 

The further experiments, such as PERK modulation with GSK2606414 or calcineurin-B (Fig 2D), and the measurement of an activating EC50-value (Fig 3C) provide evidence that the developed assay is robust and can be used to screen for PERK activators. Nevertheless, we acknowledge the reviewer’s comment and integrated the mentioned aspect in our manuscript to point out limitations of our study in line 510-517, also concerning the use of SMAD3.

Comment 2: It is unclear why SMAD3 was chosen as a substrate, as it is not a natural substrate of PERK.

Answer: Thank you for this comment. SMAD3 is used in routine analysis to measure PERK’s specific activity with a radioactive assay protocol setup by the manufacturer SignalChem. It is a cost-efficient alternative for PERK’s natural substrates, such as eIF2α or NRF2, which are both very expensive. Because we developed our assay for a future application in high-throughput screening (HTS) using large amounts of proteins, this is an important point to consider. To make it this point clear, we modified the sentence in line 333-336.

Comment 3: The use of the unnatural substrate SMAD3, of only the catalytic domain of PERK and of a reaction at room temperature instead of 37C, takes the developed assay away from the natural PERK biology. While it might still be useful to screen for PERK activators, this should be commented and discussed as a limitation of the study.

Answer: Thank you for the comment. We added this point to line 509-517 and state more clearly possible limitations of our work. 

Comment 4: The statement in the Abstract about the “…non-availability of direct PERK activators…” as well as at the beginning of the Discussion and in other parts of the text is misleading, as MK-28 has been shown to be a direct PERK activator in a cell-free assay (Ganz el al, 2019) and confirmed now in this manuscript. These statements should be removed or changed.

Answer: We changed the manuscript accordingly. To avoid misunderstandings, we changed the statement in the Abstract (line 24) from “non-availability of direct PERK activators” to “a shortage of available direct PERK activators”. In other formulation we state e.g., “lack of direct PERK activators” (line 83) or “lack of available compounds that directly activate PERK” (line 455). We believe that the term “lack” states correctly that there is not a complete absence but rather very limited availability.

Comment 5: Fig. 1A, and lines 258-9. “…cellular screening assays may identify compounds that activate the PERK pathway in an unspecific manner…” Although this could be true in general, there are ways to test the specificity to PERK, by measuring the effect of the compounds in PERK knockout cells compared to WT cells. This should be added/ commented in the figure and in the text.

Answer: Thank you for this point. We added this aspect to our manuscript in line 263, and by changing the Fig 1A legend to “by measuring distinct activated target proteins or by comparing treated PERK knockout cells with wild-type cells other pathways of the UPR can be excluded” (line 274-275).

Comment 6: Lines 259-261 “Contrariwise, cell-free screening systems, by restricting the reaction to the target protein, allow the identification of test compounds that specifically and directly interact with the PERK kinase (Fig 1B).” This statement gives the impression that direct interaction is measured in the manuscript, which is not the case. Direct action on PERK can be inferred but direct interaction has not been measured. The statement should be removed or moderated.

Answer: We agree to the reviewer’s comment and carried out the following changes to the manuscript for clarification: 

- line 269 from “directly interact with the PERK kinase” to “directly modulate PERK kinase activity”

- Fig 1B from “direct interaction” to “direct modulation”

- Fig 1B legend, line 278 from “directly interact with PERK” to “directly modulate PERK kinase activity”

- line 465 from “direct target interaction” to “direct modulation”

- line 469 from “directly interact with the target” to “directly modulate the target”.

Comment 7: Line 277 “At present, there is no commercial PERK activator assay available.” and line 458 “…there is no cell-free screening kit for PERK activators obtainable…” However, there are companies that have cell-free PERK activator assays, and one can make use of their services (see e.g. in Ganz el al 2019). This should be commented in the text.

Answer: Presently, no commercial cell-free PERK activator assay is available. However, kinase inhibitor assays can be commercially obtained. Alternatively, full kinase panels for measurement of kinase activity are available at specialized companies. Originally, Ganz et al developed MK28 from a compound termed “A4”, previously found in a virtual library screening by Wang et al (2010). After chemical modification, Ganz et al found MK28 to be more potent, and confirmed its specificity to activate PERK on a full kinase panel analysis, carried out by a company. But in conclusion, Ganz et al. didn’t use a commercially available PERK activator assay.

We do acknowledge that there are alternative methodologies, such as radioactive assays to screen for kinase modulators, as we referred in line 470, but they are intended to screen for inhibitors. We also pointed out that these methodologies are not present in all laboratorial facilities and that our developed assay has the advantage to use non-radioactive conditions with comparable sensitivity. 

Comment 8: Line 281 “…protein substrate SMAD3 from the original radioactive protocol and applying it…”. It is unclear to what original radioactive protocol the authors are referring to.

Answer: With this phrase, we refer to the radioactive protocol which is used by the manufacturer of PERK (SignalChem) to determine the specific activity of PERK as part of a routine protein quality analysis. To clarify this, we made changes to line 92-94 and line 290-291.

Comment 9: Lines 410-412 “Our findings strongly suggest that calcineurin-B place of action differs from the inhibitor GSK2606414 and activates PERK by a mechanism independent of the active site which is blocked by GSK2606414.” The authors should speculate in the text what the mechanism of activation could be, based on previous studies on the mechanisms of PERK activation.

Answer: Based on previous studies reporting PERK activation by calcineurin (Bollo et al, 2010; Chen et al, 2016), we tested and confirmed that calcineurin-B promotes PERK activity in our assay. We did not investigate the mechanism behind this activation, as it was not the scope of our project. However, based on our results, we speculated a possible mechanism of action, in lines 493: “calcineurin-B might increase PERK activity not by directly interacting with PERK’s catalytic site, but by favoring protein-protein interactions between PERK and its protein substrate”. We stated, that this speculation comes from the fact that “we observed that calcineurin-B does not favor PERK autophosphorylation and only activates PERK in the presence of a substrate protein” (line 490-491), together with “evidences of a weak PERK-calcineurin B-interaction reported by others” (line 492). Indeed, as we also noticed (line 496) “improving stability of kinase-substrate interaction, by targeting structural features prone to disruption, is a possible mechanism of action of kinase activators” (Lopez et al, 2020).

To address the reviewer´s comment, we made additional changes to line 495. 

Comment 10: Although the English is good in general, there are some mistakes in the grammar. (for example in line 373 “…when PERK reacted in the presence of calcineurin-B was neither not due to a direct…”). The text should be edited by a native English speaker.

Answer: Thank you for the comment. We corrected the mentioned error accordingly in line 385. The entire manuscript was doublechecked by a native English speaker, who we now include in the acknowledge section.

---

## [Editor Report · Decision Letter 1]

21 Mar 2023

Development of a cell-free screening assay for the identification of direct PERK activators

PONE-D-22-33273R1

Dear Dr. Costa:

We’re pleased to inform you that your manuscript has been judged scientifically suitable for publication and will be formally accepted for publication once it meets all outstanding technical requirements.

Kind regards,

Hemant K. Paudel

Academic Editor

PLOS ONE
---

## [Editor Report · Acceptance letter]

10 May 2023

PONE-D-22-33273R1 

Development of a cell-free screening assay for the identification of direct PERK activators 

Dear Dr. Höglinger:

I'm pleased to inform you that your manuscript has been deemed suitable for publication in PLOS ONE. Congratulations! Your manuscript is now with our production department. 

Kind regards, 

on behalf of

Dr. Hemant K. Paudel 

Academic Editor

PLOS ONE